# Multifunctional Design of Catalysts for Seawater Electrolysis for Hydrogen Production

**DOI:** 10.3390/ma17164057

**Published:** 2024-08-15

**Authors:** Chenmeng Cui, Haonan Zhang, Dan Wang, Jihuan Song, Ying Yang

**Affiliations:** State Key Laboratory of Heavy Oil Processing, China University of Petroleum, Beijing 102249, China; cuichenmeng2023@163.com (C.C.); haonanzhang926@163.com (H.Z.); 17852981659@163.com (D.W.); song380723659@163.com (J.S.)

**Keywords:** seawater electrolysis, hydrogen production, electrocatalysts, renewable energy, multifunctional design, nanostructures, chlorine corrosion resistance

## Abstract

Direct seawater electrolysis is a promising technology within the carbon-neutral energy framework, leveraging renewable resources such as solar, tidal, and wind energy to generate hydrogen and oxygen without competing with the demand for pure water. High-selectivity, high-efficiency, and corrosion-resistant multifunctional electrocatalysts are essential for practical applications, yet producing stable and efficient catalysts under harsh conditions remains a significant challenge. This review systematically summarizes recent advancements in advanced electrocatalysts for seawater splitting, focusing on their multifunctional designs for selectivity and chlorine corrosion resistance. We analyze the fundamental principles and mechanisms of seawater electrocatalytic reactions, discuss the challenges, and provide a detailed overview of the progress in nanostructures, alloys, multi-metallic systems, atomic dispersion, interface engineering, and functional modifications. Continuous research and innovation aim to develop efficient, eco-friendly seawater electrolysis systems, promoting hydrogen energy application, addressing efficiency and stability challenges, reducing costs, and achieving commercial viability.

## 1. Introduction

The global energy supply predominantly relies on traditional fossil fuels such as coal, crude oil, and natural gas. As energy demands increase and conventional energy sources are overexploited, environmental pollution and energy crises have become increasingly severe. To address these issues, extensive research efforts have been dedicated to exploring renewable and eco-friendly “green” technologies, such as solar energy, tidal energy, and wind energy [1,2,3,4]. Hydrogen is regarded as an ideal clean energy for the future, with diverse production methods including extraction from fossil fuels, methane reforming, and photocatalytic water splitting. Among these, water electrolysis is an effective method for producing “green” hydrogen [5,6,7,8] (Figure 1a). The generated hydrogen can serve as a clean and sustainable alternative to fossil fuels, avoiding carbon dioxide emissions from traditional fuel combustion, and is widely applied in transportation, heating, and energy-related fields. Freshwater is crucial for human life. Currently, industrial pollution, climate change, and population growth have led to shortages of clean water resources. According to global water resource statistics, 97% of the Earth’s water is seawater, with salinity ranging from 3.0% to 5.0% [9,10,11] (Figure 1b). Therefore, directly using seawater as a raw material for hydrogen production through water electrolysis can be a significant pathway for producing green hydrogen [12]. Developing efficient seawater electrolysis technology not only alleviates freshwater shortages but also promotes the hydrogen economy and the utilization of marine resources. Many governments and research institutions worldwide are actively exploring and promoting seawater electrolysis technology to achieve economically efficient, sustainable, and large-scale production of green hydrogen. These initiatives, exemplified by the European Union’s Hydrogen Strategy, Japan and South Korea’s national hydrogen strategies, and the United States Department of Energy’s Hydrogen Program, underscore the global commitment to leveraging seawater electrolysis in addressing climate change and advancing industrial decarbonization.

The electrolysis process typically includes two half-reactions: the oxygen evolution reaction (OER) at the anode and the hydrogen evolution reaction (HER) at the cathode (Figure 2). Both reactions require efficient electrocatalysts to overcome energy consumption overpotential. Extensive research efforts have led to the development of many excellent OER and HER electrocatalysts, significantly advancing the application of water splitting [14,15,16,17,18,19,20,21]. However, due to the lack of high-performance electrocatalysts specifically for the OER and HER, seawater electrolysis for hydrogen production remains highly challenging. In the HER, the inherent low conductivity of seawater results in slow HER kinetics, necessitating electrocatalysts with high conductivity [22,23]. Additionally, the presence of bacteria and microorganisms in natural seawater, as well as the formation of insoluble precipitates at the active sites of the catalyst surface, further inhibits HER performance [24,25]. Regarding the OER, the inherent four-electron process unavoidably presents high energy barriers, limiting the efficiency of seawater electrolysis [26,27,28,29,30]. Moreover, the high concentration of chloride ions in seawater (approximately 0.5 M) leads to the chlorine evolution reaction (CER) competing with the OER at relatively high potentials [31,32]. In alkaline solutions, the CER involves chlorine reacting with OH^−^ to form hypochlorite, which has an onset potential 490 millivolts higher than the OER. Therefore, highly active OER catalysts are needed to drive large current densities at overpotentials below 490 millivolts to avoid hypochlorite formation [33]. Designing and constructing highly active seawater OER and HER electrocatalysts is crucial for achieving industrial hydrogen applications. Efforts should be intensified to develop efficient seawater OER and HER electrocatalysts. Catalysts such as n-Co_3_S_4_@NF, Pt-Co-Mo, cobalt-based compounds (e.g., CoP and CoS_x_), iron-based compounds (e.g., FeNi layered double hydroxides), and molybdenum-based compounds (e.g., MoS_2_ and MoNi_4_) have demonstrated significant promise in seawater electrolysis for hydrogen production. Their high catalytic activity, stability, and cost effectiveness make them central to current research endeavors. This indicates that optimizing the structure and composition of these catalysts could further enhance the efficiency and economic viability of the electrolysis process. Currently, several review articles on seawater electrolysis catalysts exist [13,34,35], but most focus on enhancing single performance metrics or improving traditional catalysts. However, in practical applications, catalysts often need to possess multiple exceptional properties to adapt to the complex seawater environment and reaction conditions. Recognizing the importance of the multifunctional design of catalysts is key to achieving efficient and stable seawater electrolysis for hydrogen production.

This review aims to systematically summarize and discuss the latest advancements and future directions in multifunctional design strategies for enhancing the overall performance of seawater electrolysis catalysts. By comprehensively applying different strategies, significant improvements in catalyst performance can be achieved, leading to more efficient seawater splitting. Multifunctional design enables catalysts to exhibit outstanding efficiency and durability in complex seawater environments, laying a solid foundation for the practical application of sustainable energy technologies. This review provides valuable knowledge resources for researchers in the field of seawater electrolysis, indicating future research directions and challenges and promoting the development of this field.

## 2. Seawater Electrocatalytic Reaction Pathways and Fundamental Principles

### 2.1. Mechanism of Direct Seawater Electrolysis

The water electrolysis reaction requires an external stimulus, namely the potential difference between electrodes, to drive the overall cell reaction. Given that the average pH of seawater is 8.2, water electrolysis in an alkaline solution is most relevant. The specific electrode reactions are depicted in Equations (1) and (2), with the overall reaction shown in Equation (3).

The cathodic hydrogen evolution reaction (HER) is as follows:4H_2_O(l) + 4e^−^ → 2H_2_(g) + 4OH^−^(aq), E^0^ = −0.83 V vs. Standard Hydrogen Electrode (SHE)(1)

The anodic oxygen evolution reaction (OER) is as follows:4OH^−^(aq) → O_2_(g) + 2H_2_O(l) + 4e^−^, E^0^ = +0.40 V vs. SHE(2)

The overall cell reaction is as follows:2H_2_O(l) → 2H_2_(g) + O_2_(g), E^0^ = 1.23 V(3)

Under standard conditions, the minimum potential required to start splitting water into hydrogen and oxygen is 1.23 V. The standard enthalpy change for reaction (3) is +286 kJ/mol of hydrogen, with a Gibbs free energy of +238 kJ/mol of hydrogen [36]. In practical operations, the cell voltage required to drive the water electrolysis reaction is given by Equation (4):(4)$$-E_{cell}=E^{0}-\left|\eta_{a}\right|-\left|\eta_{c}\right|-IR$$
where E^0^ is the equilibrium potential difference between the two electrode reactions (1.23 V) and the remaining terms represent efficiency losses due to energy consumption that should be minimized; η_a_ and η_c_ are the overpotentials of the anode and cathode, respectively, while IR represents ohmic losses due to the current (I) passing through the cell resistance (R). High-performance electrocatalysts can minimize the η terms, while minimizing the IR term relies on good electrochemical engineering.

The reaction kinetics at the anode and cathode are significantly influenced by the electrocatalysts used. The overpotential for the oxygen evolution reaction (OER) is considerably higher than that for the hydrogen evolution reaction (HER), making it the primary source of energy loss in water electrolysis cells. Therefore, reducing the anode overpotential is a key objective for alkaline water electrolysis.

### 2.2. Mechanism of the Hydrogen Evolution Reaction

The hydrogen evolution reaction (HER) is an integral part of the overall water-splitting reaction, involving a two-electron transfer mechanism through a series of proton and electron coupling stages to generate molecular hydrogen. In alkaline media, OH^−^ ions are the primary reactants [37,38,39,40,41], and the water electrolysis pathways are illustrated in Equations (1) and (2).

To better understand the HER in direct seawater electrolysis (DSWE), it is crucial to recognize the significant role of OH^−^ ions in the cathodic reaction under alkaline conditions. Compared to acidic conditions, alkaline conditions allow the use of a broader range of non-precious metal electrocatalysts, thus reducing costs and improving feasibility. Under DSWE conditions, the cathodic reaction pathway is shown in Equation (2).

Therefore, optimizing the HER in seawater electrolysis hinges on developing efficient electrocatalysts to effectively reduce overpotential, facilitate rapid proton and electron transfer, and ultimately enhance hydrogen yield [42]. Due to the ease of accumulation of hydroxide ions in alkaline conditions, the kinetics of the OER are faster than those of the HER [40,41,42,43]. In alkaline media, the HER starts with the dissociation of water molecules to produce protons, involving three steps: the Volmer step, where water dissociates on the electrode surface to produce adsorbed hydrogen; the Heyrovsky step, where adsorbed hydrogen reacts with another water molecule to form hydrogen gas; and the Tafel step, where two adsorbed hydrogen atoms combine to produce hydrogen gas [43] (Figure 3). Due to the proton dissociation from water molecules and the formation of hydrogen intermediates (H_ad_), the HER kinetics in alkaline environments are slower compared to acidic conditions [44].

The hydrogen evolution reaction (HER) in alkaline media involves three key steps, as shown in Equations (5)–(7).

The Volmer reaction is as follows:H_2_O + e^−^ → H* + OH^−^
(5)

The Heyrovsky reaction is as follows:H* + H_2_O + e^−^ → H_2_ + OH^−^(6)

The Tafel reaction is as follows:2H_2_O + 2e^−^ → 2OH^−^ + H_2_(7)

According to Parsons’ theory, the dissociation of water and the subsequent adsorption of hydrogen atoms on the catalyst surface are relatively slow, making the early formation of hydrogen intermediates the rate-determining step (RDS) of the entire HER process [45]. The complexity of the alkaline HER, compared to the acidic HER, arises from the necessity to adjust several crucial steps, such as hydroxide adsorption, H_ad_ formation, and water dissociation, to achieve optimal performance [46,47,48].

In direct seawater electrolysis (DSWE), optimizing the HER is crucial. Although the HER in alkaline conditions is complex, the use of non-precious metal catalysts makes it a cost-effective option. Developing efficient electrocatalysts can effectively reduce overpotential, facilitate rapid proton and electron transfer, and enhance hydrogen yield. The natural electrolytes and ions present in seawater further impact the HER kinetics, making the development of electrocatalysts suited to seawater conditions a key area for future research.

### 2.3. Competition between Oxygen Evolution Reaction and Chlorine Evolution Reaction

In the direct electrolysis of seawater, the oxygen evolution reaction (OER) involves a four-electron transfer process, inevitably leading to high energy barriers and slow kinetics. Additionally, the presence of approximately 0.5 M chloride ions in seawater causes the chlorine evolution reaction (CER) to occur at the anode at higher electrochemical potentials, competing with the OER. Despite the OER having a lower thermodynamic reduction potential, its kinetics are not ideal due to the four-electron transfer process. The OER steps in alkaline electrolytes are shown in Equations (8)–(11).

In alkaline media (the symbol “*” indicates that the intermediates are adsorbed on the surface of the electrode)
OH^−^ → OH* + e^−^(8)
OH* + OH^−^ → O* + H_2_O + e^−^(9)
O* + OH^−^ → HOO* + e^−^(10)
HOO* + OH^−^ → O_2_(g) + H_2_O + e^−^(11)

These steps illustrate the specific process of the OER in alkaline electrolytes, involving multiple intermediates and electron transfers. This complexity results in high energy barriers and slow reaction kinetics, leading to significant competition between the OER and CER in seawater electrolysis.

Seawater primarily consists of water, with 3.5% by weight being salts. Among the dissolved ions, chloride ions (Cl^−^) constitute 55.04%, followed by sodium ions (Na^+^) at 30.61%. This is why an approximately 0.5 M NaCl solution is typically used to simulate seawater. Other ions such as sulfate (7.76%), magnesium (3.69%), calcium (1.16%), potassium (1.10%), bicarbonate (0.41%), bromide (0.19%), borate (0.07%), and strontium (0.04%) also interfere with reactions at the electrodes. However, due to the high concentration of chloride ions, their effect is the primary concern, as they compete with the desired oxygen evolution reaction. The competition between chlorine and oxygen evolution is represented by the Pourbaix diagram (Figure 4).

The following is seen at pH 0:2Cl^−^ → Cl_2_ + 2e^−^, E^0^ = 1.36 V vs. RHE (12)

The following is seen at pH 14:Cl^−^ + 2OH^−^ → ClO^−^ + H_2_O + 2e^−^, E^0^ = 1.72 V vs. RHE(13)

The electrochemical oxidation of chloride ions is complex, with reactions varying depending on pH, temperature, and applied potential. Figure 4 shows the potential–pH regions where the OER and chlorine oxidation reactions are thermodynamically possible. For simplicity, we consider a standard temperature (298 K) and a chloride ion concentration of 0.5 M (C_T,Cl_), typical of seawater chloride levels. The green line represents the thermodynamic equilibrium before water splits into oxygen. If the electrode potential is more positive than the green line, the OER is thermodynamically favorable. However, if the potential exceeds the pink line (Equation (13)), chlorine oxidation is more favorable. Therefore, as highlighted by the blue region, there is an approximate 480 mV potential window in alkaline environments where oxygen evolution can occur without oxidation. The red line indicates the competition between chlorine oxidation (ClOR) and chlorine evolution reactions (Equation (12)) [49].

Understanding and optimizing the competition between the oxygen evolution reaction (OER) and the chlorine evolution reaction (CER) is crucial in direct seawater electrolysis. The high concentration of chloride ions not only affects electrolysis efficiency but may also lead to the formation of byproducts. Therefore, developing highly selective electrocatalysts that can effectively facilitate the OER while avoiding the CER is a significant goal for seawater electrolysis.

## 3. Challenges in Seawater Splitting

### 3.1. Overall Challenges

Electrolysis in seawater becomes complicated due to ion interference and pH fluctuations at the cathode, hindering the reaction. In unbuffered seawater, the kinetics of the hydrogen evolution reaction (HER) are slow, potentially causing local pH changes at the cathode surface, leading to the precipitation of dissolved ions. When pH fluctuations exceed 9.5, it may result in catalyst degradation and the formation of magnesium hydroxide and calcium hydroxide precipitates, blocking active sites and reducing electrode activity. Issues such as salt deposition, microorganisms, bacteria, and small particles also present persistent challenges. However, these can be mitigated by introducing turbulence, supporting electrolytes, selecting appropriate catalysts, and optimizing current density [50].

Despite extensive studies on salts in seawater, data on microorganisms and their effects on direct seawater electrolysis (DSWE) are limited. Existing microbial studies lack comprehensive investigations, leaving this area relatively unexplored. Comparative studies of water splitting in simulated and actual seawater have shown a decrease in current density or an increase in overpotential in real seawater, but the specific chemical reasons for performance degradation remain unexplained. In seawater electrolysis, biofouling is a major challenge, leading to the blockage of active sites, membrane issues, and reduced equipment lifespan [51]. Although measures have been taken to alleviate this problem, implementing direct seawater applications may introduce complexities, such as the need for multilayer electrode coatings, potentially affecting catalytic performance [52,53].

Overall, addressing these challenges requires optimizing catalyst selection, controlling electrolysis conditions, and conducting in-depth studies on microbial impacts to improve the efficiency and feasibility of seawater electrolysis.

### 3.2. Challenges with OER Catalysts

To efficiently produce hydrogen through direct seawater electrolysis (DSWE), a highly selective oxygen evolution reaction (OER) catalyst is needed. Studies have shown that increasing the pH of seawater by adding potassium hydroxide (KOH) can enhance the selectivity of the anodic reaction and expand the potential window for the OER (Figure 5) [33,54,55,56].

Under highly alkaline conditions, the products of the chlorine evolution reaction (CER) shift from hypochlorous acid to hypochlorite, known as the chlorine oxidation reaction (ClOR). Under standard conditions (25 °C and 0.5 M), hypochlorite is kinetically more favorable than the OER, but the OER remains thermodynamically more advantageous [57]. The redox potential of standard hypochlorite oxidation is strongly influenced by pH; the pH-dependent slope gradient matches that of the OER potential in the Pourbaix diagram. In alkaline solutions, the formation potential of hypochlorite is approximately 480 mV higher than the OER. Therefore, if electrocatalytic oxidation can achieve complete seawater electrolysis below 1.72 V RHE and the OER overpotential in alkaline electrolytes is less than 480 mV, hypochlorite formation is theoretically suppressed in this potential region, allowing for nearly 100% selectivity [58]. This is the basis for measuring OER electrocatalyst activity in DSWE literature relative to the theoretical threshold of 480 mV.

Thus, a design criterion for selective OER control is proposed: under conditions of pH greater than 7.5, the reaction condition is as shown in Equation (14):η_OER_ < 480 mV, at pH > 7.5(14)

Figure 5 illustrates that for the 100% selective OER region, the overpotential needs to be below 480 mV, and the pH must be at least 7.5. Lowering the pH reduces the potential required for the ClOR to compete with the OER. Therefore, the anode catalyst must be highly selective to minimize the formation of highly corrosive hypochlorite during seawater electrolysis.

Overall, optimizing oxygen evolution reaction (OER) catalysts to achieve high selectivity and low overpotential remains one of the key challenges in the seawater electrolysis process. By rationally designing catalysts and controlling electrolysis conditions, it is possible to enhance electrolysis efficiency and reduce byproduct formation, thereby achieving more efficient hydrogen production.

## 4. Progress in Multifunctional Design of Electrocatalysts for Seawater Hydrogen Production

### 4.1. Structure and Material Design Strategies for HER Catalysts

Currently, seawater electrolysis is widely recognized as an economically efficient method to produce hydrogen energy. Hydrogen evolution reaction (HER) catalysts play a crucial role in renewable energy conversion and storage. Recent studies have identified contaminants in seawater as the primary reason for the low efficiency and instability of the cathodic HER, as these contaminants can corrode and block the active sites of conductive electrodes, affecting the reaction. To achieve efficient and stable HER performance in various electrolytes and corrosive environments, researchers have developed a variety of multifunctional design strategies. These strategies mainly include the design of nanostructures and surface modifications, alloying and multi-metal catalyst construction, atomic dispersion, and interface engineering. These designs aim to enhance multiple properties of the catalyst through multifunctional strategies. For instance, nanostructure design can increase the specific surface area of the catalyst, improving the exposure rate of active sites; surface modifications help enhance the catalyst’s corrosion resistance and conductivity. Alloying and multi-metal catalyst construction improve the selectivity and stability of the catalyst by adjusting the electronic structure and optimizing the reaction pathways. Strategies such as atomic dispersion and interface engineering increase the utilization of active sites and improve the reaction selectivity and long-term stability by introducing highly dispersed active atoms and precisely designed interface structures into the catalyst. These designs not only enhance the activity and selectivity of the catalyst but also significantly improve its corrosion resistance and long-term stability.

#### 4.1.1. Nanostructure Design and Surface Modification Strategies

Nanostructure design has a significant effect on increasing the specific surface area of catalysts and the exposure of active sites. By increasing the specific surface area of the catalyst, nanostructures can provide more active sites, effectively promoting electrochemical reactions. For example, Xu et al. synthesized nanoporous C-Co_2_P materials through an electrochemical etching method (Figure 6a). In a 1.0 M KOH solution, this material required only a low overpotential of 30 mV at a current density of 10 mA cm^−1^ (Figure 6b), demonstrating its excellent performance in the HER [59]. The study showed that carbon doping significantly promoted HER activity (Figure 6c). This nanoporous structure significantly enhances HER performance by increasing active sites and optimizing electron transport pathways. Zhao et al. prepared Co-doped VS_2_ nanosheets, introducing abundant S defects and active edge sites (Figure 6d), effectively improving catalytic reaction efficiency. This catalyst exhibited excellent HER performance in alkaline seawater with a Tafel slope of 214 mV dec^−1^ and stable operation for 12 h [60]. Yu et al. constructed a nanostructured NiCoN|Ni_x_P|NiCoN catalyst, which, due to its large surface area and high conductivity, significantly increased the number of active sites and electron transport efficiency. This catalyst required only 165 mV of overpotential at a current density of 10 mA cm^−2^, showing outstanding selectivity and corrosion resistance [61].

#### 4.1.2. Alloying and Multi-Metallic Composite Design

Alloy and multi-metallic catalysts, by optimizing electronic interactions between metals, can effectively enhance electrocatalytic activity and provide superior stability during electrolysis. The design of these catalysts aims to leverage the synergistic effects of different metals to modulate the electronic structure, thereby improving catalytic performance. For instance, Yuan et al. prepared a NiMo thin-film catalyst via electrodeposition (Figure 7a). By regulating the oxidation states of Ni and Mo, they achieved a low overpotential of 31.8 mV in a 1 M KOH + 0.5 M NaCl solution (Figure 7b,c) [62]. This alloying strategy optimized the electronic interaction between Ni and Mo, enhancing HER activity. Similarly, Ros et al. developed a carbon-supported Ni-Mo-Fe electrocatalyst, which exhibited excellent corrosion resistance and stability in alkaline seawater due to alloying and reconstruction [63]. This multi-metallic composite introduced Fe to improve the electronic structure, thereby increasing the corrosion resistance of the catalyst and electrochemical stability. Moreover, Wang et al. synthesized a Co-doped Fe_2_P transition metal phosphide catalyst via a hydrothermal method. XPS measurements revealed the presence of Fe, Co, O, and P atoms on the surface (Figure 7d). The shift of Fe 2p3/2 and Fe 2p1/2 peaks towards lower binding energies indicated electronic interactions between Fe and Co due to Co doping (Figure 7e). This catalyst achieved overpotentials of 138 and 221 mV at a current density of 100 mA cm^−2^ in 1 M KOH and simulated seawater, respectively [64]. The Co doping optimized the electronic structure of Fe_2_P, improved its electron distribution and Fermi level, reduced the activation energy of catalytic sites, and enhanced its bifunctional catalytic performance in chloride-containing environments.

#### 4.1.3. Single-Atom Catalysis and Interface Engineering Optimization

Through atomic dispersion and interface engineering, the atomic structure and interface characteristics of catalysts can be precisely regulated, optimizing their electrochemical performance at the molecular level. The application of these strategies is significant in enhancing the multifunctionality and stability of catalysts. Zang et al. designed a Ni-N_3_ catalyst with a triple nitrogen coordination structure (Figure 8a) [65], which displayed high-efficiency HER activity in alkaline electrolytes without any activity decay over 14 h. To further investigate the detailed structure of Ni-SA/NC, Fourier-transform extended X-ray absorption fine structure (FT-EXAFS) spectroscopy was employed. The spectra showed a peak at 1.40 Å, attributed to Ni-N/C coordination, with no detectable Ni-Ni peak, indicating the absence of Ni nanoparticles (Figure 8b). EXAFS measurements confirmed that the coordination environment and chemical state of the Ni single atoms remained unchanged after cycling, indicating strong interactions between the active Ni atoms and the N-doped carbon substrate (Figure 8c). This single-atom catalysis strategy, by finely tuning the coordination environment of Ni atoms, enhances the stability and activity of the catalyst. Wu et al. developed a multifunctional catalytic interface combining MXene, bimetallic carbide, and hybrid carbon. An X-ray diffraction (XRD) analysis revealed that Co_x_Mo_2−x_C/NC had a structure similar to Co_0.31_Mo_1.69_C/MXene/NC (Figure 8d). This catalyst demonstrated HER activity comparable to commercial Pt/C in 1.0 M KOH and 0.5 M H_2_SO_4_ and outperformed Pt/C in the pH range of 2.2–11.2 (Figure 8e) [66]. This interface engineering strategy optimized electron transport and reaction activity by integrating the advantages of different materials. Zhang et al. prepared a CMO film by anchoring molybdenum–oxygen functional groups on a Cu substrate through in situ electrochemical reduction, significantly enhancing the selectivity and corrosion resistance of the catalyst [67]. This interface engineering approach optimized the electronic structure by anchoring molybdenum–oxygen functional groups on the Cu substrate, improving the selectivity of the catalyst and corrosion resistance.

The multifunctional design of catalysts is crucial for enhancing the performance of hydrogen evolution reaction (HER) catalysts in seawater electrolysis. The three main strategies currently include nanostructuring and surface modification, alloying and multi-metallic catalysts, and atomic dispersion and interface engineering. These designs significantly improve the catalyst’s activity, selectivity, and corrosion resistance by increasing the surface area, optimizing the electronic structure, and enhancing interface stability. These advancements provide valuable insights for developing efficient and stable HER catalysts, promoting the progress of renewable energy technologies.

### 4.2. Optimization Strategies for OER Catalysts in Seawater Electrolysis

The oxygen evolution reaction (OER) is a crucial half-reaction in water splitting, widely applied in the electrochemical conversion and storage of renewable energy. Direct seawater electrolysis for hydrogen production offers advantages such as low investment cost, fewer engineering challenges, and smaller equipment footprint. However, the presence of a high concentration of chloride ions (Cl^−^) in seawater can lead to competing chlorine evolution reactions (ClERs), reducing the Faradaic efficiency of the OER. Additionally, abundant calcium and magnesium ions in seawater can deposit on cathode catalysts, causing deactivation. Therefore, an ideal OER catalyst must exhibit high activity, good conductivity, corrosion resistance, and long-term stability in harsh environments. Current research employs various strategies, including nanostructuring, alloy and heterostructure design, functional modification, and interface engineering, to enhance OER catalyst performance. These multifunctional strategies not only focus on improving individual properties but also aim for the synergistic optimization of multiple properties. For instance, nanostructuring increases the surface area and active sites, significantly enhancing the catalyst’s reactivity. The introduction of alloys and heterostructures improves selectivity and stability by modulating the electronic structure and catalytic mechanisms. Functional modifications and interface engineering enhance corrosion resistance and long-term stability by introducing specific functional groups and optimizing interface structures.

#### 4.2.1. Nanostructuring and Two-Dimensional Material Optimization

Nanostructures and two-dimensional materials significantly enhance the electrochemical performance and long-term stability of catalysts by increasing the surface area and exposing more active sites. These material design strategies play a crucial role in improving the selectivity and stability of catalysts. For example, Cui et al. [68] decorated nickel foam with Fe-Ni(OH)_2_, forming Ni_3_S_2_ nanorods, which significantly improved the selectivity of the catalyst and its stability. This nanostructure enhanced the electrochemical reaction efficiency by providing more active sites and optimizing transport pathways. Similarly, Song et al. [69] synthesized a MoS_2_-(FeNi)_9_S_8_ nanostructure (Figure 9a), exhibiting excellent performance in alkaline seawater. This nanostructure combined MoS_2_ with (FeNi)_9_S_8_, increasing the catalyst’s surface area and active sites, effectively enhancing its catalytic activity in alkaline seawater (Figure 9b). Furthermore, Yu et al. [70] synthesized S-(Ni,Fe)OOH on nickel foam using a simple method, forming a porous structure and enhancing stability. X-ray photoelectron spectroscopy (XPS) measurements confirmed the presence of the elements Ni, Fe, O, and S in the S-(Ni,Fe)OOH layer on the engineered nickel foam surface (Figure 9c). This porous structure increased the surface area and exposed more active sites, improving the catalyst’s electrochemical stability and reaction efficiency (Figure 9d).

#### 4.2.2. Alloy and Heterostructure Design

Designing alloys and heterostructures can optimize the electronic interactions and structural stability between metals, thus enhancing the performance of oxygen evolution reaction (OER) catalysts. These methods significantly improve the corrosion resistance and electrochemical stability of catalysts. For instance, Li et al. [71] designed a NiFeB/NiFeB_x_/NiFe multilayer catalyst that introduced boron species to enhance corrosion resistance. This multilayer structure optimized the electronic configuration and anti-corrosion properties of the catalyst through the incorporation of boron. Similarly, Cong et al. [72] deposited CeO_2_ nanoparticles on the surface of Co_0.4_Ni_1.6_P nanowires (Figure 10a), enhancing the durability and stability of the catalyst. The deposition of CeO_2_ nanoparticles provided additional active sites and optimized interfacial electronic transport, thereby improving the long-term performance of the catalyst (Figure 10b,c). Haq et al. [73] prepared Gd-Mn_3_O_4_@CuO-Cu(OH)_2_ by adjusting oxygen vacancies and hierarchical structures, significantly improving charge transfer efficiency. This heterostructure optimized the charge transfer pathways, thereby enhancing the catalyst’s electrochemical performance. Wu et al. [74] designed a CoP_x_||CoP_x_@FeOOH core–shell structure catalyst, exhibiting high selectivity and corrosion resistance in simulated seawater. The FeOOH coating on the CoP_x_ surface provided additional active sites and optimized interfacial electronic transport, significantly enhancing selectivity and anti-corrosion properties. Wu et al. [75] also prepared Ni_2_P-Fe_2_P nanosheets via phosphorization (Figure 10d), demonstrating high activity and superior transport efficiency (Figure 10e). The nanosheet structure optimized the electronic configuration and transport pathways through phosphorization, thus improving the activity and transport efficiency of the catalyst (Figure 10f).

#### 4.2.3. Functional Modification and Interface Engineering

Functional modification and interface engineering can introduce structural defects, dope different elements, or construct multifunctional interfaces, further optimizing the electrochemical performance of catalysts. These strategies play a crucial role in enhancing the activity, stability, and conductivity of catalysts. For example, Li et al. [76] attached Fe(Cr)OOH and Fe_3_O_4_ to a nickel foam substrate, enhancing OER activity and stability. This interface engineering optimized the electronic transport pathways and active site distribution, thereby improving the catalyst’s oxygen evolution reaction performance. Chen et al. [77] formed a hollow MIL-88(FeCoNi) structure through polypyrrole and tannic acid modification (Figure 11a), exposing numerous active sites and enhancing conductivity. This functional modification created a hollow structure through the synergistic action of polypyrrole and tannic acid, increasing the exposure of active sites and improving conductivity (Figure 11b,c). Ma et al. [78] prepared NiFe-LDH nanoarrays/nickel foam by adding SO_4_^2−^ ions to alkaline seawater (Figure 11d), significantly extending the anode catalyst’s lifespan. This interface engineering introduced SO_4_^2−^ ions into the electrolyte (Figure 11e), optimizing the catalyst’s surface chemical environment and enhancing its durability and long-term stability.

#### 4.2.4. Stability Enhancement and Electrochemical Durability Design

Researchers have successfully prepared various catalysts with special structures and high performance through electrochemical corrosion and other chemical methods. These methods significantly improve the stability and electrochemical durability of catalysts. For instance, Cui et al. [68] decorated Ni foam with Fe-Ni(OH)_2_ via an electrochemical method, forming Ni_3_S_2_ nanorods (Figure 12a), showing excellent performance in seawater. X-ray photoelectron spectroscopy (XPS) measurements confirmed the presence of Ni, Fe, O, and S atoms on the Fe-Ni(OH)2/Ni3S2@NF surface (Figure 12b). Electrochemical measurements demonstrated that the Fe-Ni(OH)2/Ni3S2@NF catalyst exhibited superior OER activity compared to both Ni3S2@NF and Ni foam (Figure 12c). This nanorod structure optimized the catalyst’s configuration and electronic transport pathways, thereby enhancing its oxygen evolution reaction performance in seawater. Gao et al. [79] prepared Ni/α-Ni(OH)_2_ heterostructures through chemical and electrochemical corrosion methods, demonstrating significant durability. This heterostructure optimized the catalyst’s surface structure and stability through the combination of chemical and electrochemical corrosion methods, enhancing its electrochemical durability.

The analysis indicates that various strategies have their own merits in enhancing the performance of oxygen evolution reaction (OER) catalysts. The optimization of nanostructures and two-dimensional materials primarily boosts performance by increasing the specific surface area and exposing more active sites. Alloys and heterostructures enhance stability through the optimization of interactions between metals. Functional modifications and interface engineering introduce defects and different elements to achieve performance improvements. Additionally, electrochemical corrosion and chemical methods provide effective means for preparing catalysts with special structures. Future research should integrate these strategies to develop more efficient and stable OER catalysts that meet practical application requirements.

## 5. Conclusions and Outlook

The world’s oceans offer an almost inexhaustible supply of seawater, providing a rich and economical source of hydrogen. Direct seawater electrolysis is a promising method for the commercial and large-scale production of hydrogen. However, the unavoidable presence of chloride ions in seawater leads to severe corrosion and oxidation, which significantly hinders the application of this technology. Over the past decades, extensive efforts have been dedicated to the material modification and structural optimization of electrocatalysts, aiming to develop the most efficient electrochemical systems and highly stable catalysts to make hydrogen production commercially viable. This paper comprehensively reviews the current knowledge on seawater electrolysis catalysts. Most reported catalysts lack the necessary catalytic activity and stability for practical applications, leading to slow progress in this field. Therefore, seawater splitting remains in its infancy. We believe that seawater electrolysis for hydrogen production is a challenging task, particularly when using catalysts primarily composed of low-cost components. The design and preparation of highly active, multifunctional electrocatalysts that can suppress pollutants is crucial.

For direct seawater electrolysis, the hydrogen evolution reaction (HER) produces a large amount of surface OH^−^, leading to the in situ precipitation of Ca^2+^ and Mg^2+^, which adhere to the cathode and block active sites. While accelerated electrolyte flow can partially mitigate cathode contamination, it still does not meet the requirements for long-term, stable operation. Therefore, designing inherently anti-fouling cathodes is the best solution for future research. In alkaline seawater electrolysis, further improvement of the catalytic activity of the cathode is necessary to reduce energy consumption and achieve stable operation under fluctuating energy supplies. The electrocatalysts used in seawater electrolysis must also have high stability to resist oxidation by Cl_2_, ClO_2_, and other similar compounds. The oxygen evolution reaction (OER) catalysts for seawater electrolysis need to have low overpotential, long-term stability, and resistance to Cl^−^ corrosion. Therefore, the development of nanostructured catalysts is essential to ensure low energy consumption, high impurity tolerance, high stability, low cost, and low corrosiveness. To gain a deeper understanding of the reaction processes in seawater splitting, it is critical to develop advanced characterization methods, such as in situ microscopy, spectroscopy, and chromatography. These methods will help expand innovative catalysts and electrolyzers to meet the demands of industrial-scale hydrogen production. Seawater electrolysis for hydrogen production is a field on the verge of significant breakthroughs. With a detailed and in-depth evaluation of materials and further research and improvement of electrocatalyst systems, breakthroughs are expected in the future.

This review discusses highly selective multifunctional electrocatalysts for the HER and OER in seawater splitting and introduces multifunctional strategies to improve the stability and corrosion resistance of seawater electrolysis systems. Current research in this field is still limited, and further studies should focus on enhancing the stability, selectivity, and corrosion resistance of seawater electrolysis systems, as these are prerequisites for future industrialization. In conclusion, the development of multifunctional electrocatalysts with high stability, selectivity, and excellent resistance to chlorine corrosion for direct seawater lysis is urgently needed.

## Figures and Tables

**Figure 1 materials-17-04057-f001:**
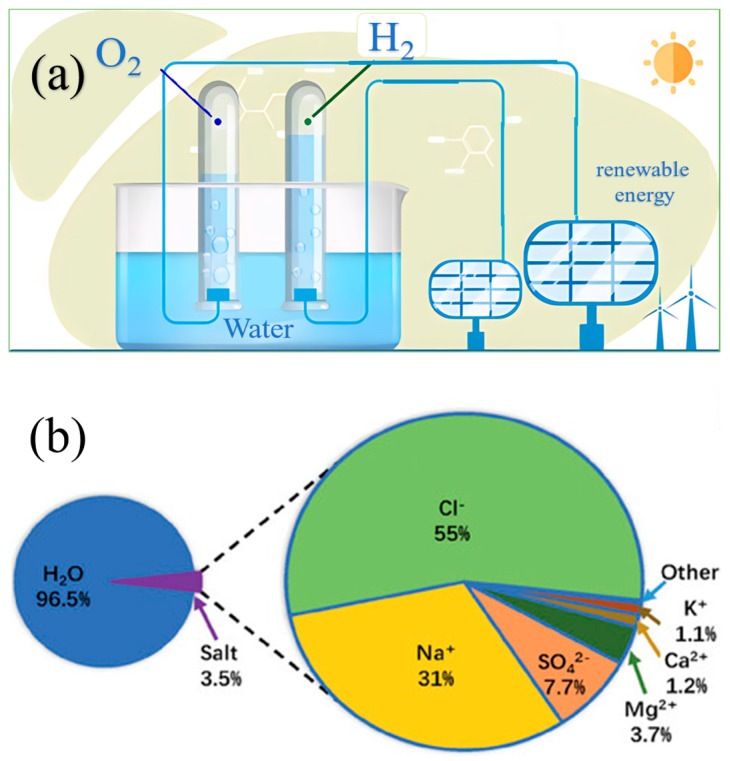
(**a**) Schematic illustration of the catalytic mechanism in water electrolysis. From the Internet. (**b**) Pie chart of the mass fraction of various components in seawater. From Yu and Liu (2023) with permission from Wiley [13].

**Figure 2 materials-17-04057-f002:**
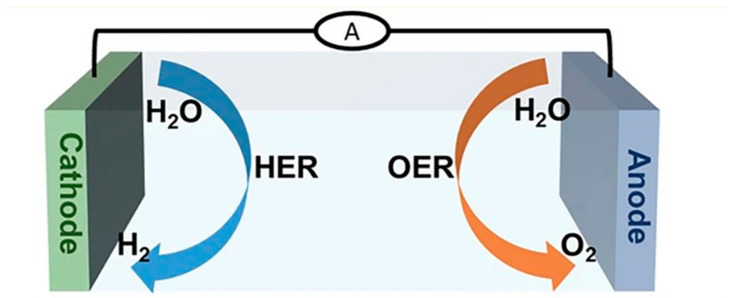
Schematic diagram of the principles of the oxygen evolution reaction (OER) and hydrogen evolution reaction (HER). From Gao et al. (2024) with permission from Springer [34].

**Figure 3 materials-17-04057-f003:**
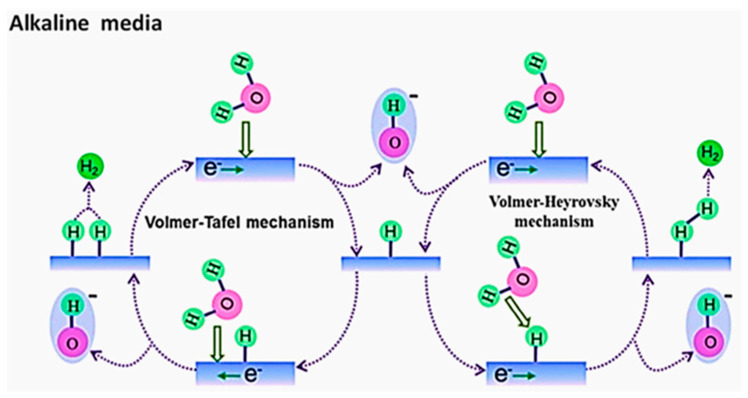
Illustration of the HER process under alkaline conditions. From Aldosari et al. (2023) with permission from Elsevier [35].

**Figure 4 materials-17-04057-f004:**
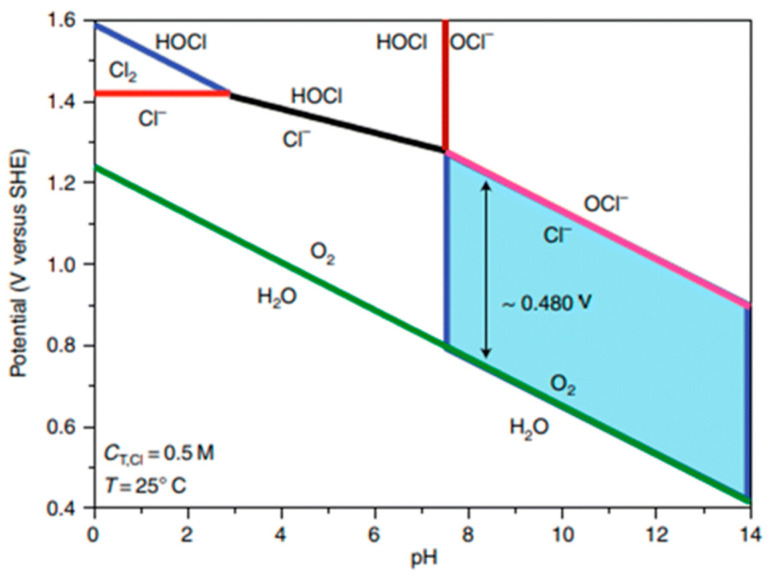
Pourbaix diagram of saline electrolyte. From Dresp et al. (2019) with permission from ACS Publications [8].

**Figure 5 materials-17-04057-f005:**
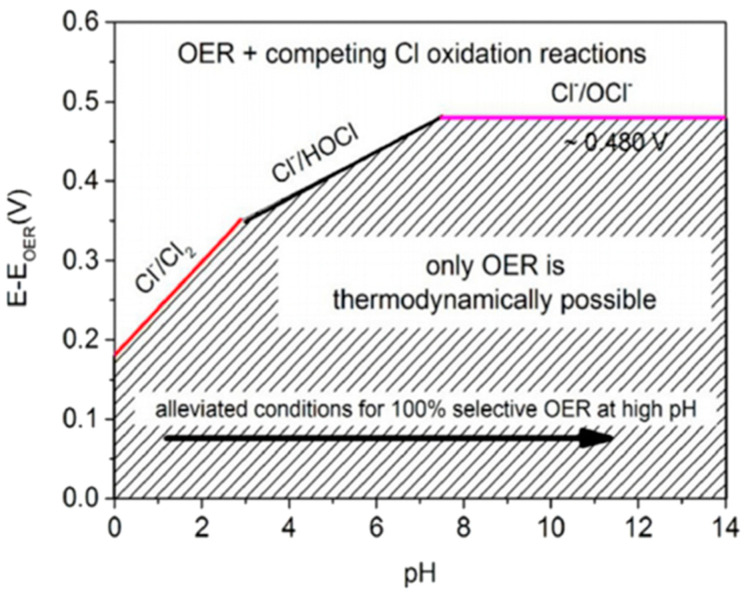
Maximum permissible overpotential for OER electrocatalysts. From Dionigi et al. (2016) with permission from Wiley [56].

**Figure 6 materials-17-04057-f006:**
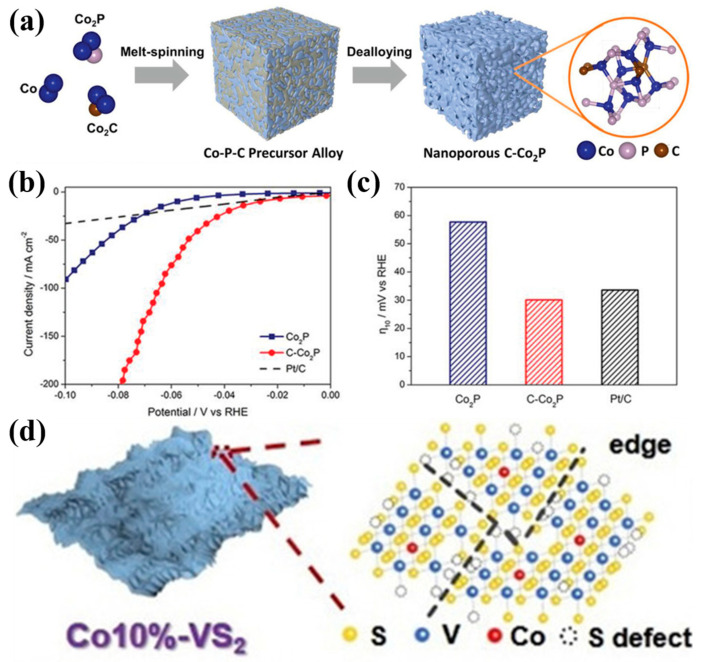
(**a**) Schematic illustration of the preparation of nanoporous electrocatalyst C-Co_2_P. (**b**) Polarization curves of C-Co_2_P, Co_2_P, and Pt/C catalysts in 1 M KOH. (**c**) Comparison of overpotentials. (**d**) Schematic representation of the composition of Co10%-VS_2_. (**a**–**c**) From Xu et al. (2021) with permission from Wiley [59]. (**d**) From Zhao et al. (2021) with permission from Wiley [60].

**Figure 7 materials-17-04057-f007:**
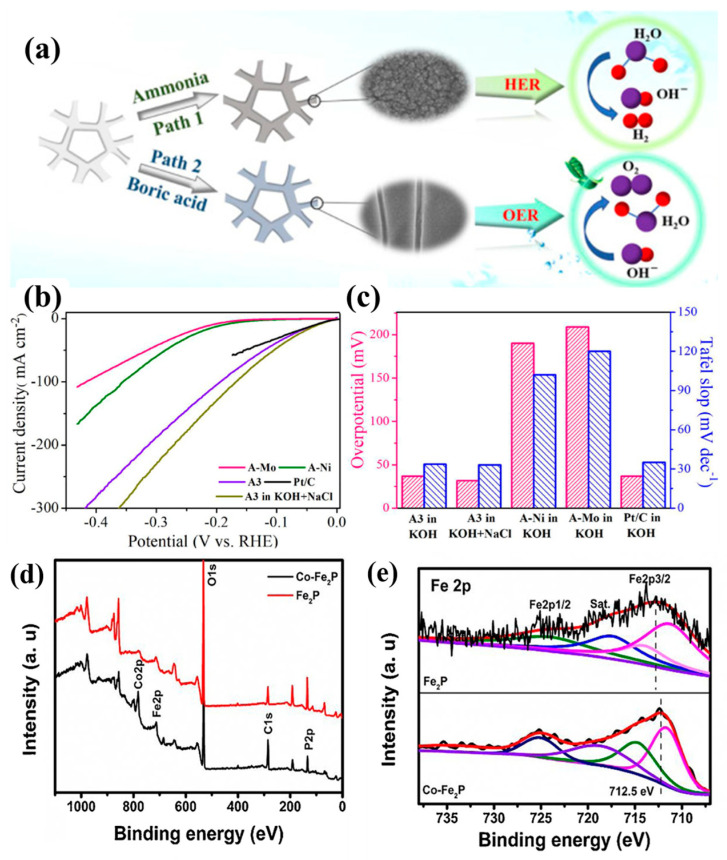
(**a**) Schematic illustration of the preparation of nickel–molybdenum films as HER/OER catalysts. (**b**) Polarization curves of A3, A-Ni, A-Mo, 20% Pt/C tested in 1 M KOH, and A3 tested in 1 M KOH + 0.5 M NaCl. (**c**) Corresponding bar charts of overpotentials and Tafel slopes. (**d**) XPS analysis of Fe_2_P and Co-Fe_2_P electrocatalysts. (**e**) Fe 2p XPS spectra of Fe_2_P and Co-Fe_2_P electrocatalysts. (**a**–**c**) From Yuan et al. (2021) with permission from Elsevier [62]. (**d**,**e**) From Wang et al. (2021) with permission from Elsevier [64].

**Figure 8 materials-17-04057-f008:**
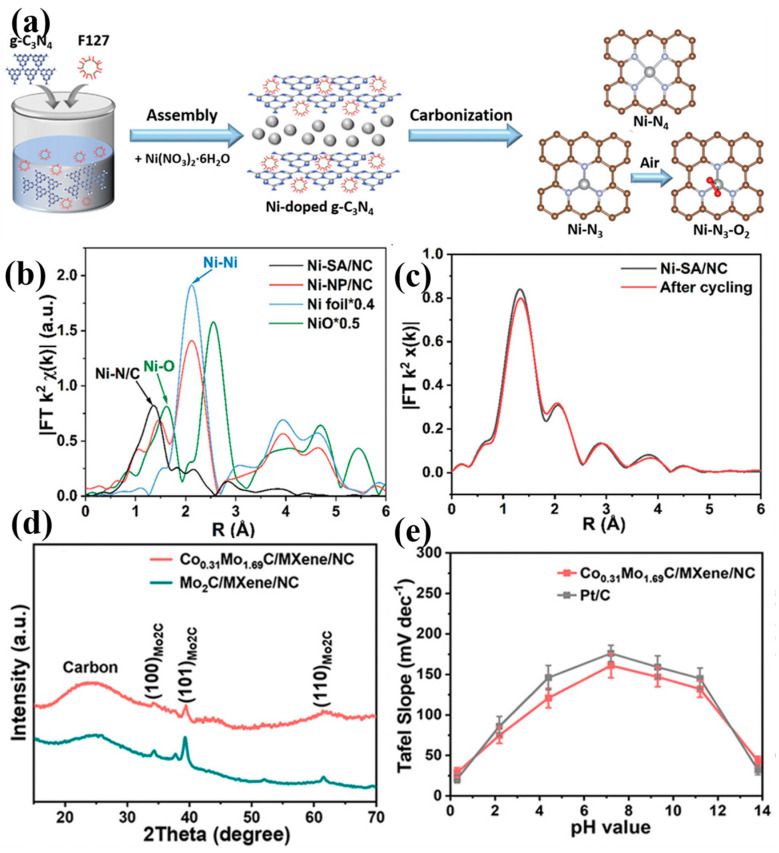
(**a**) Schematic illustration of the fabrication process of Ni-SA/NC. (**b**) Fourier-transform extended X-ray absorption fine structure (FT-EXAFS) spectra. (**c**) EXAFS curves of the prepared Ni-SA/NC before and after 5000 cycles. (**d**) XRD patterns of Co_0.31_Mo_1.69_C/MXene/NC and Mo_2_C/MXene/NC. (**e**) Comparison of the Tafel slopes for Co_0.31_Mo_1.69_C/MXene/NC and 20% Pt/C over a pH range of 0.3 to 13.8. (**a**–**c**) From Zang et al. (2021) with permission from Wiley [65]. (**d**,**e**) From Wu et al. (2019) with permission from Wiley [66].

**Figure 9 materials-17-04057-f009:**
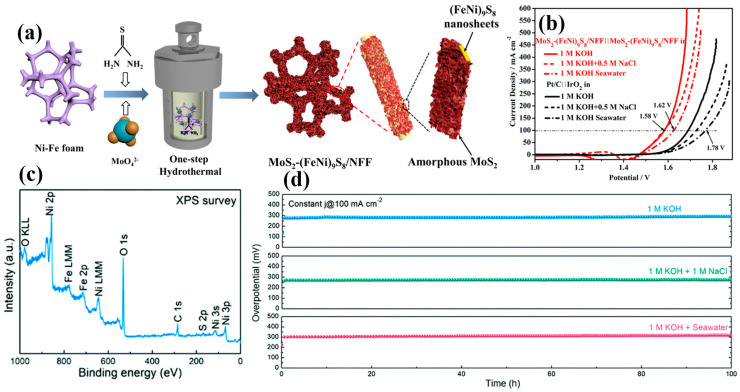
(**a**) Schematic illustration of MoS_2_-(FeNi)_9_S_8_/NFF with composite nanosheets. (**b**) Polarization curves of MoS_2_-(FeNi)_9_S_8_/NFF and Pt/C||IrO_2_ electrolyzers in three different electrolytes. (**c**) XPS spectra of S-(Ni,Fe)OOH. (**d**) Long-term stability tests of the S-(Ni,Fe)OOH electrode at a constant current density of 100 mA cm^−2^ in various electrolytes. (**a**,**b**) From Song et al. (2022) with permission from ACS Publications [69]. (**c**,**d**) From Yu et al. (2020) with permission from RSC [70].

**Figure 10 materials-17-04057-f010:**
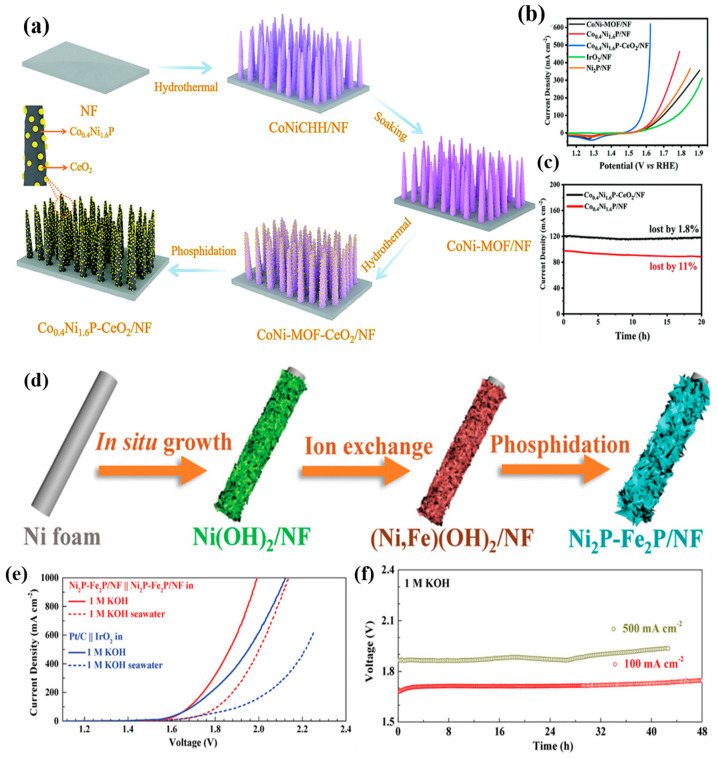
(**a**) Schematic illustration of the preparation of Co_0.4_Ni_1.6_P–CeO_2_/NF. (**b**) LSV curves of various electrocatalysts. (**c**) Current density vs. time curves for Co_0.4_Ni_1.6_P–CeO_2_/NF and Co_0.4_Ni_1.6_P/NF. (**d**) Synthesis of Ni_2_P-Fe_2_P/NF electrocatalyst via a three-step in situ growth, ion-exchange, and phosphorization method. (**e**) Overall water/seawater-splitting performance of Ni_2_P-Fe_2_P/NF and Pt/C||IrO_2_ in 1 M KOH and 1 M KOH seawater. (**f**) Chronopotentiometric curves of Ni_2_P-Fe_2_P/NF in 1 M KOH at constant current densities of 100 and 500 mA cm^−2^. (**a**–**c**) From Cong et al. (2022) with permission from RSC [72]. (**d**–**f**) From Wu et al. (2021) with permission from Wiley [75].

**Figure 11 materials-17-04057-f011:**
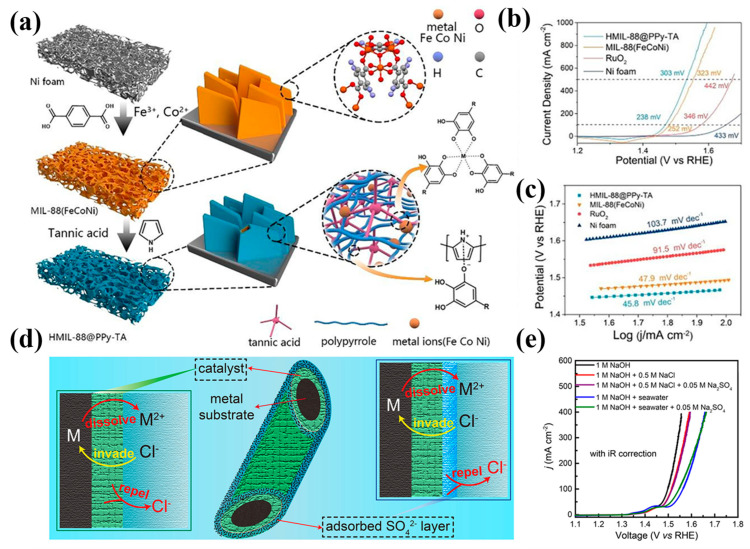
(**a**) Schematic illustration of the preparation of HMIL-88@PPy-TA on nickel foam. (**b**) CV curves of HMIL-88@PPy-TA, MIL-88 (FeCoNi), RuO_2_, and nickel foam in an electrolyte solution of 1 M KOH. (**c**) Tafel plots of HMIL-88@PPy-TA, MIL-88 (FeCoNi), RuO_2_, and nickel foam in 1 M KOH. (**d**) Optimization of catalysts (**left**) and electrolytes (**right**) to protect metal substrates from Cl^−^ corrosion. (**e**) LSV curves for the OER of NiFe-LDH nanorod arrays/NF in electrolytes with and without SO_4_^2−^. (**a**–**c**) From Chen et al. (2022) with permission from Elsevier [77]. (**d**,**e**) From Ma et al. (2021) with permission from Wiley [78].

**Figure 12 materials-17-04057-f012:**
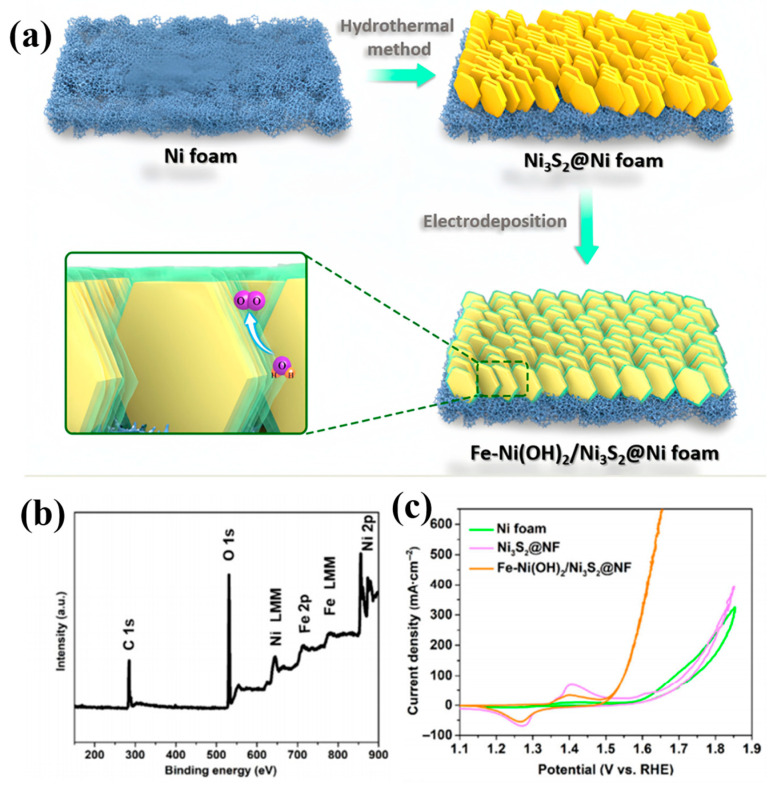
(**a**) Schematic illustration of the synthesis of layered-edge Fe-Ni(OH)_2_/Ni_3_S_2_ nanorod arrays. (**b**) XPS spectra of Fe-Ni(OH)_2_/Ni_3_S_2_@NF. (**c**) CV curves of Fe-Ni(OH)_2_/Ni_3_S_2_@NF, Ni_3_S_2_@NF, and nickel foam in 1 M KOH/0.5 M NaCl at a scan rate of 5 mV·s^−1^. (**a**–**c**) From Cui et al. (2021) with permission from Springer [68].

## Data Availability

No new data were created or analyzed in this study.

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
