# Peer review of "Multifunctional Design of Catalysts for Seawater Electrolysis for Hydrogen Production"

_materials, 2024, doi:10.3390/ma17164057_

Round 1

Reviewer 1 Report

Comments and Suggestions for Authors

This is a well-written review manuscript with an easy-to-follow approach. My only major comment on this is that a mention of the target for hydrogen production from seawater electrolysis that various governments are striving for and what is the best catalyst available among the plethora of reported catalysts will make this manuscript very strong.

Some minor comments are below

Page 2, line 65 -67 – authors claim there are several review articles available on sea water electrolysis but no references provided. Need to provide a few.

Page 3, Equation4 – the equation is not clear with the sign placement.

Equation 6 and equation 7 are identical while eq 2 and 3 are nearly the same as equation 5 and 6.

Page 5, line 156, Had formation where ad has to be subscript.

equation 17 has a mandarin symbol which I don’t know what it is for

Page 15, line 488 – exhibited better OER activity – it is not clear better than what?

Line 539 – in-s microscopy?

Reviewer 2 Report

Comments and Suggestions for Authors

This review manuscript deals with the recent achievements in the advanced electrocatalysts for seawater splitting, focusing on their multifunctional designs for selectivity and chlorine corrosion resistance. In addition, a detailed overview of the fundamental principles and mechanisms of seawater electrocatalytic reactions, the challenges, the progress in nanostructures, alloys, multi-metallic systems, atomic dispersion, interface engineering, and functional modifications was also made. Since the heterogeneous catalysis have high impact on the environmentally friendly processes including hydrogen energy applications, this overview can be meaningful for the future developments on this field.

The manuscript is fairly well written and the results are clearly presented. However, some minor revisions should be made before publication:

1) p. 2 line 44 It seems that all figures are reproduction from the appropriate references, but it is not mentioned in the captures of these figures. You should give them in all cases.

2) p. 4 lines 123 and 129 The equations 6 and 7 are the same, thus you should check their numbering.

3) p. 4 line 142 The size of Figure 3 should be increased, because it is difficult to recognize its content. There is a similar situation with Figures 6–8, 11 and 12. Please, modify them.

4) p. 5 line 173 The authors use asterisk (*) in the reaction equations 11–14, but it is not clear what its meaning. Does it refer to a radical? Please, clarify this.

5) pp. 17–20 Style of the references does not meet the requirements of journal Materials. For example, in case of “… Chimia 2015, 69, ...” you should use the following form: Chimia 2015, 69,. Practically, the all references contain some mistakes. Please, check and modify them carefully.

Comments on the Quality of English Language

The English also needs slight improvements. There are some, typical mistakes:

p. 3 line 94 and elsewhere „… E0 …”  instead of   E0

p. 4 line 138 „Because of ...  instead of  Due to

p. 5 line 156 „… Had …”  instead of  had

p. 7 line 258 „… 在 …”  instead of  ???

p. 9 line 328 „… the catalyst's corrosion resistance …  instead of  the corrosion resistance of the catalyst

p. 11 line 368 and elsewhere „… catalyst's selectivity…  instead of  the selectivity of catalyst

p. 13 line 448  „… the catalyst's activity and transport efficiency …  instead of  the activity and transport efficiency of the catalyst
